

# Surface-subsurface interaction analysis and the influence of precipitation spatial variability on a lowland mesoscale catchment

Faisal Sardar[1], Muhammad Haris Ali[1,2], Ioana Popescu[1,2], Andreja Jonoski[1], Schalk Jan van Andel[1], Claudia Bertinia[1]

[1]Department of Hydroinformatics and Socio-Technical Innovation, IHE Delft Institute for Water Education, P.O. Box 3015, 2601 DA Delft, The Netherlands
[2]Water Resources Section, Delft University of Technology, 2628 CD Delft, The Netherlands

*Correspondence to*: Muhammad Haris Ali (h.ali@un-ihe.org)

**Abstract.** The hydrology of the catchments is primarily shaped by the intricate and dynamic interactions between surface water and groundwater. This is particularly evident in lowland catchments, where these interactions assume a complex nature. This study investigated the complex interaction between surface water and groundwater in the transboundary catchment Aa of Weerijs, shared by the Netherlands and Belgium. A hydrological model, MIKE SHE coupled with MIKE 11, was calibrated and validated over twelve years using streamflow, groundwater levels, and evapotranspiration data. The model performance was analyzed using model efficiency parameters i.e., correlation coefficient (R) and Nash-Sutcliffe Efficiency coefficient (NSE). The model performed well, with satisfactory simulations of streamflow, groundwater levels, and evapotranspiration dynamics. Groundwater levels rose in winter and declined from April to September due to increased evapotranspiration in summer. Precipitation drove the water balance, with 60% lost through evapotranspiration. Base flow from subsurface drainage networks significantly contributed to river water. Spatial variability in precipitation minimally impacted streamflow but caused localized fluctuations in groundwater levels. Higher spatial resolution precipitation data led to fluctuations due to local recharge points, yet overall catchment hydrology was unaffected. The findings highlight the importance of surface water-groundwater interactions in lowland catchments. The developed model provides insights for water resource planning and climate change adaptation in the catchment.

**Keywords.** Aa of Weerijs; MIKE SHE; mesoscale catchments; lowland; Precipitation; Surface & Groundwater interactions

## 1 Introduction

The understanding of interaction between surface water and groundwater is essential for managing hydro systems (Jeannot et al., 2019). By using a multi-scale approach that combines various techniques from different disciplines, we can reduce uncertainties and gain a comprehensive understanding of the dynamics between groundwater and surface water (Ntona et al., 2022). Surface water and groundwater, the main components of the hydrological system, are often analyzed separately. However, their interaction is vital for hydrosystem functionality. This interaction occurs through gaining stream and losing stream, depending on hydraulic head. Seasonal precipitation variations and events can impact groundwater tables and stream



stages, thereby influencing the exchange flow (Kalbus et al., 2006). The dynamic interplay between surface water and groundwater is intricately influenced by various factors, including the unique characteristics of the catchment area and the temporal variations in climate conditions. To deepen our comprehension of this hydrological relationship, researchers have employed two primary approaches: empirical field-based techniques and numerical modelling (Nippgen et al., 2011; Woods, 2005).

Traditionally, empirical field-based approaches have been employed to assess and quantify exchange flow in stream-aquifer systems. The literature offers several techniques, each with varying applicability depending on the study's focus and spatiotemporal scope (Kalbus et al., 2006). The interaction within the lowland stream-aquifer system was analysed by Krause et al. (2012) utilizing the heat tracer method to assess exchange fluxes, and Poulsen et al. (2015) employed isotopic tracer methods to qualify groundwater fluxes to the stream. Landon et al. (2001) on the other hand, employed hydraulic conductivity to develop an appropriate technique for sandy bed rivers. These studies demonstrate the diverse range of methods used to investigate and understand the interaction between surface water and groundwater.

Although these approaches have been widely used, particularly at a small scale and under current conditions, their applicability may be limited when considering future scenarios involving changes in land use and climate conditions (Yang et al., 2017). In contrast, numerical modelling is a valuable tool for assessing and quantifying this interaction at a regional scale. Unlike traditional field-based methods, numerical modelling provides a structured approach to analyse and comprehend the intricate dynamics of water interactions (Zhou et al., 2022). In literature, two types of models, the conceptual model and the physical-based model are being used for surface and groundwater interaction. Conceptual models use the movement of water between different conceptual stores to represent surface-water groundwater interaction (Herron and Croke, 2009; Kazumba et al., 2008). While the physical-based distributed model uses the hydraulic properties of the water and soil to represent the interaction (Yi-Luo, 2011).

Physical-based fully distributed hydrological models, such as MIKE SHE, offer a means to simulate the complex interactions between hydrological processes within catchments. These models rely on accurately representing various input parameters to ensure realistic results. However, the reliability of hydrological models greatly depends on the availability of high-quality hydrological datasets, encompassing essential variables like precipitation, wind speed, temperature, and solar radiation, which are crucial for precise streamflow simulation (Price et al., 2014).

In lowland catchments, where the interplay between surface water and groundwater becomes complex, the influence of climate drivers, particularly precipitation, becomes more pronounced due to shallow groundwater levels (GWL). Consequently, acquiring detailed and accurate catchment water balance information becomes increasingly important to effectively manage water resources (Walsum et al., 2001; Waseem et al., 2020).

Precipitation is one of the most critical hydrological parameters that require accurate representation in fully distributed hydrological models. It serves as the primary driver of the hydrological cycle, and its spatial and temporal variability greatly affects the estimation of water balance components such as runoff, infiltration, and groundwater recharge. That makes the accurate representation of precipitation essential for predicting properly hydrological processes in a catchment (Fu et al.,





2011). Numerous studies have emphasized the importance of accurately representing precipitation in hydrological modelling, including the utilization of remote sensing data (Ali et al., 2023; Cui et al., 2019; Hiep, et al., 2018)

The impact of rain gauge station density on hydrological simulated runoff has been investigated, indicating that a decrease in density leads to lower-quality outcomes in hydrological models (Bárdossy and Das, 2008). However, factors such as precipitation type, catchment size, behaviour, model type, and watershed characteristics also contribute to the results

(Bredesen and Brown, 2019; Price et al., 2014). Wiebe and Rudolph (2020) studied the influence of spatial density of rain gauges on numerical estimation of groundwater recharge for the Alder Creek watershed, Waterloo Canada and concluded that the recharge is quite sensitive to rain gauge network. However, the impact of spatial variability in precipitation data on the interaction between surface-subsurface hydrology has gained little attention and may influence catchment water budget estimation.

Previous research in this field has predominantly concentrated on either surface water or groundwater dynamics in isolation, often disregarding the intricate interplay between these vital components within lowland catchments. Additionally, the current body of literature offers limited insights into the influence of climate drivers, particularly precipitation patterns, on these interactions, particularly in the context of shallow groundwater systems. This research centers its attention on the Aa of Weerijs catchment, also characterized as a lowland mesoscale catchment grappling with drought issues attributable to

climate change. This study aims to enhance our understanding of the intricate interplay between surface water and groundwater, facilitating the successful implementation of adaptation strategies to mitigate the effects of climate change. Therefore, this paper aims to present, firstly, insights into the hydrological behavior and complex interaction between surface water and groundwater for a mesoscale lowland catchment. Secondly, it aims to explore the impact of spatial variability in precipitation data on the interaction between surface water and groundwater, as well as on the overall hydrology of the

catchment. A physically-based fully distributed hydrological model MIKE SHE coupled with MIKE 11, has been used and applied to the small lowland catchment located in the Netherlands, the Aa of Weerijs.

After this introduction, the paper provides an overview of the study area. It then presents the research materials and methods used in the study. The results obtained from the research are presented, followed by a comprehensive discussion of the findings. Finally, the paper concludes with a summary of the key findings and their implications.

**2 Study area**



The research focuses on investigating the hydrological dynamics of the Aa of Weerijs catchment, a transboundary catchment shared by Belgium and the Netherlands, as depicted in Fig. 1. This catchment spans an area of 346 km², with 199 km² located in Belgium and 147 km² in the Netherlands. The Aa of Weerijs River originates in the Flanders region of Belgium near Brecht and flows towards Breda in the Netherlands, eventually converging into small canals.

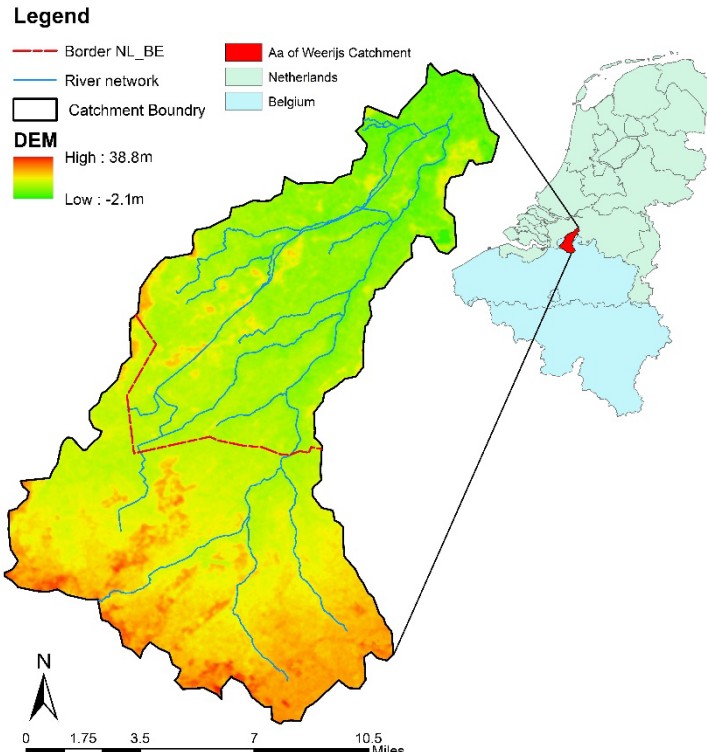

**Figure 1: Aa of Weerijs catchment**

The historical background of the catchment reveals its initial state as a vast swamp, which was later reclaimed for agricultural purposes in the 14th century. To manage water resources effectively, various water regulation structures were installed on the river, particularly in the late 1960s. These structures play an important role in regulating the flow and water levels within the catchment. The climate in the region is characterized as temperate maritime, with precipitation occurring 100 fairly evenly throughout the year. The average annual precipitation ranges between 700 mm to 900 mm. Rainfall patterns are influenced by the North Atlantic climate, resulting in relatively mild winters and cool summers (Lu et al., 2022).

The hydrology of the Aa of Weerijs catchment is influenced by various factors, including topography, land use, soil type, and climate. The main river, Aa of Weerijs, maintains a relatively constant discharge and is regulated by controlled structures. It receives contributions from several tributaries, including Kleine Beek, Bijloop, and Turfvaart. The discharge 105 hydrograph at the outlet point of the catchment exhibits variations throughout the year, with an average annual peak discharge of 23 $m^3 s^{-1}$. Geologically, the catchment primarily consists of Quaternary deposits, including sand, gravel, and





clay. Sandy soils are prevalent in the Dutch part of the catchment, with sand ridges found in areas such as Zundert and Rijsbergen. The catchment's geological composition significantly impacts its hydrological behaviour, influencing factors such as groundwater recharge, surface water runoff, and erosion. Currently, the dominant land use in the catchment is

agriculture, with grassland covering the majority of the area (Beers et al., 2018).

## 3 Materials and Methods

Numerical modelling has been extensively employed by researchers to investigate the surface-subsurface hydrology of catchments. However, its applicability in lowland catchments, where surface-subsurface interactions are more intense and complex due to shallow GWL remains limited. MIKE SHE is a powerful tool for distributed hydrological modelling,

providing a comprehensive framework for simulating all the major components of the hydrological cycle and integrating detailed spatial information into the model.

This study aims to setup a distributed hydrological model by coupling MIKE SHE and MIKE 11. The focus is on investigating the intricate interaction between surface water and groundwater in a mesoscale lowland catchment. Additionally, the study examines the impact of spatial variability in precipitation on catchment hydrology. The methodology

employed to achieve these objectives is described in the following section, which begins with a concise introduction to the study area.

### 3.1 Model Setup and Input Data

The hydrological analysis of the study area i.e., Aa of Weerijs utilized the MIKE SHE modelling system. Derived from the SHE model, MIKE SHE is a deterministic, distributed, and physically-based hydrological model. It incorporates the

complete land phase of the hydrological cycle, encompassing both surface water and sub-surface water (Abbott et al., 1986; Refsgaard and Abbott, 1996). However, model input requirements may vary depending on the complexity of the model setup and the goals of the simulation. The model resolution was set at 500 m x 500 m, and it utilized the digital elevation model EU-DEM version 1.1 (Copernicus n.d.). The simulation period covered the timeframe from 15-09-2009 to 31-12-2016.

To incorporate the river network and hydraulic structures, the MIKE SHE model was coupled with MIKE 11. The river

cross-section data for the Dutch part of the catchment were obtained from the SOBEK model provided by Waterschap Brabantse Delta, while for the Belgian part, the cross-sectional data was derived from the catchment's topography. A total of twenty-nine hydraulic structures were included in the river network, exclusively located in the Dutch part of the catchment out of which 22 are fixed weirs and 07 are automated weirs. Regulatory effect was considered on the 07 automated weir to maintain the upstream water level as per the operation guidelines of Waterschap Brabantse Delta.

Climate data, a key parameter for hydrological analysis, was obtained from radar data for the Netherlands and three rain gauge stations. A data shift issue was discovered during pre-processing with the Dutch rain gauge stations which was corrected with time-weighted factors. The main land use types observed in the catchment is complex cultivation patterns also



referred as permanent grass land as per National Hydrologic Instrumentation-NHI sub-report on crop characteristics (NHI, 2007) and five soil textures cover the catchment area with sandy loam and loam being the dominant as shown in the

supplementary material (Fig. S1). The model's saturated zone consists of a single 80 m deep aquifer layer. To define the movement of water in and out of the catchment, the outer boundary condition was divided into twenty-nine segments, using an average of seven years of GWL (2010-2016). Summary of the input data set and its sources are summarized in Table 1.

**Table 1: Summary of model input datasets**

| Data Set | Source | Resolution | Data Availability |
|---|---|---|---|
| Digital elevation model (DEM) | EU-DEM version 1.1 | 25m X 25m | - |
| Precipitation<br>(i)   Rain gauge stations<br>(ii)  Radar Data | (Flanders n.d.), (Meteobase, n.d.)<br>(Meteobase, n.d.) | -<br>1000m X 1000m | 2009-2021<br>2009-2021 |
| Potential evapotranspiration | Gilze-Rijen weather station Netherlands (Makkink) | Uniform | 2009-2021 |
| Actual evapotranspiration | Satellite data (Meteobase, n.d.) | 100m X 100 m | 2012-2016 |
| Land use map | Corine land cover (CLC-2018) (Copernicus n.d.) | 100m X 100m | - |
| Land use parameters | NHI sub-report crop characteristics | - | - |
| Overland flow computation | Manning M<br>Papaioannou et al. (2018). | 500m X 500m | - |
| Unsaturated flow computation | Richard's Eq. (Rosetta program – H1) (Schaap, et al. 2001) | - | - |
| Soil map | European soil data centre database (ESDAC) | 500m X 500m | - |
| Soil hydraulic properties | Rosetta program, USA | - | - |
| Saturated hydraulic conductivity<br>(i)  Netherlands<br>(ii) Belgium | REGIS II V2.2 model, Netherlands Iterative interpolations | 500m X 500m | 2009-2021 |
| Ground water level<br>(i)   Netherlands<br>(ii)  Belgium | WBD water board & (DINOloket n.d.) Databank Ondergrond (Flanders n.d.) | 500m X 500m | 2009-2021 |
| Subsurface drainage component | WBD water board | 500m X 500m | - |

**3.2 Model performance analysis**

The model performance was evaluated at multi-site through multi-variable. The variables considered are discharge, groundwater levels, and actual evapotranspiration. Discharge results were compared at three specific locations in the Dutch part of the catchment, namely the outlet, the middle of the catchment, and the NL-BE border. For groundwater levels, a total of 13 locations were selected based on the availability of data. Further, for actual evapotranspiration, 13 points were selected

to cover each land-use type in the catchment. These points were distributed throughout the catchment, as illustrated in Fig. 2.





For the evaluation of model performance two commonly used performance metrics, the correlation coefficient (R) and the Nash-Sutcliffe Efficiency coefficient (NSE) were used. In addition to that, Kling Gupta Efficiency (KGE) is used as an evaluation indicator specifically for GWL model performance. The KGE metric is used to address some of the limitations of the NSE, which can place greater emphasis on peak values. The KGE metric provides a comprehensive evaluation of the

model's ability to capture the observed variability, correlation, and mean values of the hydrological system being studied (El-Nasr et al., 2005).

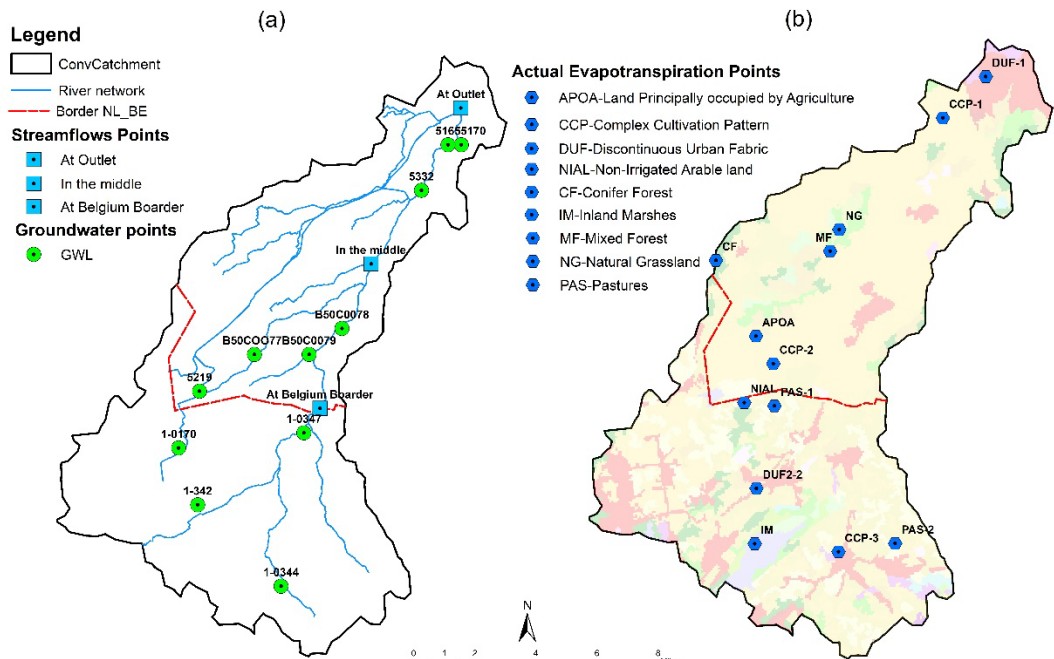

**Figure 2: Location of points for model performance evaluation (a) streamflow & GWL (b) actual evapotranspiration**

### 3.3 Calibration and validation

The calibration of hydrological models plays a crucial role in ensuring accurate model predictions. By comparing the model's predictions to observed outcomes, researchers can identify areas where the model deviates from reality and make necessary adjustments. This iterative process helps improve the model's accuracy and reduces the risk of drawing incorrect conclusions from the data (Rajib et al., 2016).

For the application of a distributed hydrological model in watershed management, detailed model calibration is essential to

obtain consistent and reliable results. Previous studies (Ambroise et al., 1995; Refsgaard, 1997) have highlighted that calibrating a distributed hydrological model solely against single-point hydrological parameters may not yield satisfactory results for the entire catchment. The criteria to calibrate and evaluate the hydrological models play an important role in its performance (Cinkus et al., 2022). To achieve better performance, it is recommended to employ multiple variables and multiple site calibration strategies in distributed hydrological modelling.



The use of multiple sites for calibration is a valuable strategy for reducing uncertainties and improving the overall quality of model simulations (Bekele and Nicklow, 2007; Chiang et al., 2014). However, the effectiveness of this approach is influenced by the distribution and location of observation points across the catchment area (Migliaccio and Chaubey, 2007). Vázquez et al. (2008) employed a multi-framework approach to calibrate a distributed hydrological model and suggested that the multi-criteria calibration approach enhances the model's ability to replicate the physical phenomena within the

catchment. Similarly, Wang et al. (2012) compared single-site and multi-site calibration approaches in a study of a catchment in north China and found that the multi-site calibration strategy was necessary to capture the spatial variability of hydrological processes.

In this research, a comprehensive approach involving multiple sites and variables was employed for model calibration. The calibration process began with a sensitivity analysis to identify the most influential parameters. Weighted objective functions

were then developed, considering appropriate weights for multiple variables and locations. This step ensured that the calibration process accounted for the spatially distributed behaviour of the model. To achieve a fully distributed behaviour of the model, calibration parameters were uniformly and spatially distributed. This approach considered the heterogeneity of the catchment and aimed to capture the spatial variability of model parameters across the study area. By distributing the calibration parameters, the model was better able to represent the complex hydrological processes occurring within the

catchment.

### 3.3.1 Weighted objective function

To account for multiple variables and different locations during the calibration of the distributed hydrological model, a weighted objective function was utilized. This objective function incorporated the correlation coefficient and Nash-Sutcliffe Efficiency (NSE) as evaluation metrics and assigned specific weights to each variable and location. The weighted objective

functions are presented below in Eq. 1 and 2:

$$MNSE = w_1 (NSE\text{-}Q) + w_2(NSE\text{-}E) + w_3(NSE\text{-}G) \tag{1}$$

$$MR = w_1(R\text{-}Q) + w_2(R\text{-}E) + w_3(R\text{-}G) \tag{2}$$

Where

MNSE= Mean weighted Nash-Sutcliffe Efficiency, MR = Mean weighted correlation coefficient, w = Weights

Assigning appropriate weights is crucial for an effective calibration process. Different approaches have been used by researchers to determine the weights, but a consensus on the optimal weight selection process has not been reached. Some studies, such as Rode et al. (2007) and Abbaspour et al. (2007), assigned weights inversely proportional to the variance of the observed data. Rientjes, et al. (2013) used weights inversely proportional to the coefficient of variation. However, due to the lack of clear justification for selecting appropriate weights, assigning equal weights has been commonly adopted in

calibration studies (Franco and Bonumá, 2017; Rajib et al., 2016).

In this study, equal weights of 0.333 were assigned to each variable type to ensure equal consideration. However, different weights are assigned to observation locations where the variable is being measured. Important locations received higher





weights. For streamflow, a higher weight of 0.4 was assigned to the outlet point due to its importance in water management and policy-making. The other two points received equal weights of 0.3. For groundwater levels, all points were considered equally important, and thus an equal weight of 0.08 was assigned to each point. In the case of actual evapotranspiration, weights were assigned based on the abundance of land use types in the catchment, with higher weights ranging from 0.1 to 0.05 given to the most prevalent land use types. The detailed weights assigned for each variable and locations are presented in supplementary material (Table S1).

### 3.3.2 Multi-site and multi-variate calibration

The distributed hydrological model used in this research considered multiple variables at different locations to evaluate its performance. Manual calibration was chosen over automated calibration as it provided valuable insights into catchment behaviour by allowing for the examination of changes in calibration parameters. This approach facilitated a better understanding of the model's deficiencies and shortcomings.

A sensitivity analysis was performed to determine the most influential parameters affecting catchment hydrology and the interaction between surface water and groundwater. During the sensitivity analysis, each parameter was adjusted individually while keeping the others constant, allowing for a comprehensive understanding of their impact on catchment hydrology. The parameters considered in the sensitivity analysis, along with their associated model components, are contained in supplementary material (Table S2). After the sensitivity analysis, the fully distributed hydrological model MIKE SHE coupled with MIKE 11 was calibrated by using the mean weighted objective function which considers all three variables distributed throughout the catchment. The parameter considered in the calibration process and their optimal values are mentioned below in Table 2.

**Table 2: Model parameters for calibration along with value ranges**

| Parameter | Distribution | Initial Value | Range | Final Value |
|---|---|---|---|---|
| Manning's roughness coefficient (M) | Uniform | 30 | 20-50 | 30 |
| Leakage Coefficient | Uniform | $4.0e^{-7}$ | $4.0e^{-5}$–$4.0e^{-7}$ | $4.0e^{-7}$ |
| Specific storage | Uniform | 0.0001 | 0.0001-0.0002 | 0.0001 |
| Specific yield | Uniform | 0.2 | 0.1-0.3 | 0.2 |
| Horizontal hydraulic conductivity | Spatially Distributed | $2.8e^{-5}$–$6.3e^{-5}$ | $2.5e^{-5}$–$10e^{-5}$ | $2.6e^{-5}$–$7.2e^{-5}$ |
| Vertical hydraulic conductivity | Uniform | $4.36e^{-5}$ | $4.36e^{-5}$– $4.36e^{-7}$ | $4.36e^{-5}$ |
| Time Constant | Spatially Distributed | $2.3e^{-7}$–$3.44e^{-7}$ | $2.0e^{-7}$–$5.0e^{-7}$ | $2.0e^{-7}$–$5.0e^{-7}$ |





The calibration process for the horizontal hydraulic conductivity parameter involved deriving a range of values from the
National hydrogeological model of the Netherlands REGIS II v2.2 (DINO n.d.) An iterative approach was employed to
develop a distributed map by starting with the minimum value and gradually increasing it until the optimal model
performance was achieved. This iterative process allowed for the identification of the hydraulic conductivity values that best
represented the hydrological behaviour of the catchment. Similarly, an iterative interpolation technique using Inverse
Distance Weighting (IDW) was employed to determine the optimal value set for the drainage time constant.

### 3.3.3 Model validation

Model validation is a crucial step in hydrological modelling as it ensures the reliability and accuracy of the model's
predictions. In this study, a five-year time frame from 2017-2021 was considered for model validation, based on data
availability. The variables considered for model validation are the same as that of model calibration. Their specific details
and data sources, are contained in supplementary material (Table S3). To assess the performance of the validated model,
efficiency parameters (R and NSE) were used.

### 3.4 Impact of precipitation spatial variability

This research paper also aimed to assess the influence of spatial variability in precipitation on catchment hydrology. This
study employed the final calibrated model to examine the response of the catchment under different representations of
precipitation. Precipitation data from radar measurements in the Netherlands and daily corrected time series data from three
rain gauge stations were utilized to represent precipitation ranging from uniform to fully spatially distributed patterns.
Three distinct scenarios were examined in the model using the corrected rain gauge data and radar data set. The first scenario
involved the application of the Theisen polygon method, utilizing data from three rain gauge stations (Ginneken, Zundert,
and Leonhout). In the second scenario, the Inverse Distance Weighting (IDW) interpolation technique was employed,
utilizing daily time series data from the rain gauge stations. The third scenario incorporated radar data for the Dutch portion
of the catchment, while interpolated data for Belgium was considered. The interpolation was carried out by incorporating the
three locations in the Netherlands' radar data set along with data from the Leonhout rain gauge station located in Belgium, as
depicted in Fig. 3.



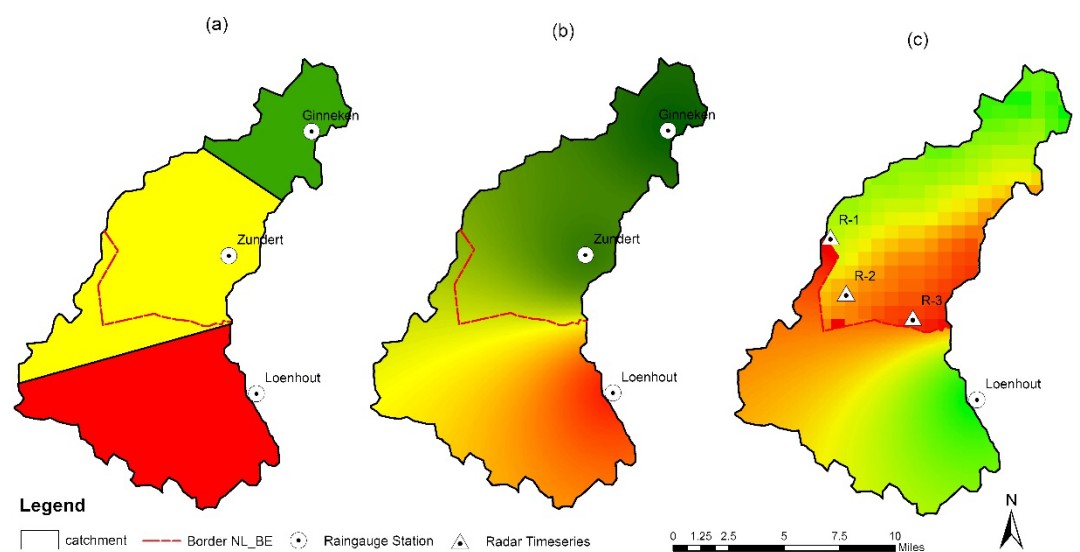

**Figure 3: Precipitation representations (a) Theisen polygon (b) Inverse distance weighting IDW (c) Radar Data**

**4 Results & Discussion**

**4.1 Calibration and validation**

The calibration process focused on adjusting calibration parameters to improve the model's performance. While exploring various channel flow and saturated zone parameters did not yield significant improvements, fine-tuning the horizontal hydraulic conductivity and drainage time constant showed promising results. It is important to note that streamflow and

groundwater levels are interconnected through the drainage network. Hence, a reduction in drainage levels diverts more water into the stream, resulting in an improved model performance for streamflow. However, this adjustment adversely affected the model's performance in predicting groundwater levels (GWL). To address the interconnections between streamflow and groundwater, a multisite and multivariate calibration strategy using weighted objective functions (MR and MNSE) played a significant role. The results, including parameter variations and the weighted mean objective functions, can

be found in Table 3.

**Table 3: Model results in terms of the mean weighted objective function for the calibration period (2010-2016)**

| Parameter | Distribution | Variation | | Model Performance | |
| --- | --- | --- | --- | --- | --- |
| | | | | MR | MNSE |
| Hz. Hydraulic conductivity | Spatial Distributed | $KH_{min}$ | $2.9e^{-5}$ - $5.8e^{-5}$ | 0.840 | 0.547 |
| | | $KH_{min}$-$I_{10\%}$ | $3.2e^{-5}$ -$6.4e^{-5}$ | 0.841 | 0.550 |
| | | $KH_{min}$-$I_{20\%}$ | $3.5e^{-5}$ - $7.0e^{-5}$ | 0.842 | 0.550 |
| | | $KH_{min}$-$I_{25\%}$ | $3.6e^{-5}$ - $7.2e^{-5}$ | 0.843 | 0.551 |





| | | | | |
|---|---|---|---|---|
| | $KH_{min}$-$I_{30\%}$ | $3.8e^{-5}$ - $7.5e^{-5}$ | 0.843 | 0.550 |
| | $KH_{min}$-$I_{40\%}$ | $4.1e^{-5}$ - $8.1e^{-5}$ | 0.843 | 0.547 |
| | $KH_{min}$-$I_{50\%}$ | $4.4e^{-5}$ - $8.7e^{-5}$ | 0.845 | 0.546 |
| | $KH_{max}$ | $5.8e^{-5}$ - $11.6e^{-5}$ | 0.850 | 0.524 |
| Drainage time constant | Uniform | $2.0e^{-07}$ | 0.841 | 0.541 |
| | | $3.0e^{-07}$ | 0.841 | 0.546 |
| | | $4.0e^{-07}$ | 0.836 | 0.534 |
| | | $5.0e^{-07}$ | 0.834 | 0.524 |
| | Spatial Distributed | $2.0e^{-7}$ - $5.0e^{-7}$ | 0.843 | 0.555 |

Model optimal performance was achieved with spatially distributed horizontal hydraulic conductivity ($KH_{min}$-$I_{25\%}$) in the range of $3.6e^{-5}$–$7.2e^{-5}$ and drainage time constant distributed values of $2.0e^{-7}$–$5.0e^{-7}$. The detailed model results against these

weighted objective functions along with model validation results can be seen in supplementary material (Table S4).

## 4.2 Catchment Hydrology

### 4.2.1 Surface Water Dynamics

The outcome of the simulated streamflows demonstrated satisfactory performance, as evidenced by the model efficiency parameters R and NSE, with values ranging from 0.76–0.90 and 0.571–0.780, respectively. In general, the model tended to

underestimate streamflows between 5 and 20 $m^3s^{-1}$, while overestimated low flows (below 2 $m^3s^{-1}$) which can be seen in Fig. S2 provided in supplementary material. The primary explanation for this discrepancy may be attributed to the subsurface drainage network, which serves as the main contributor to streamflows. When the groundwater level (GWL) remains beneath the drainage level, the subsurface drainage system becomes inactive, resulting in reduced contributions to the river and, consequently, lower streamflow peaks. Secondly the coarse model resolution of 500 x 500 m may not be representing the

actual field condition. Additional factors that could contribute to this observed behaviour may include the biased saturated zone (SZ) boundary conditions, hydraulic conductivity, and drainage time constant.

### 4.2.2 Groundwater dynamics

A good correlation exists between the simulated and observed groundwater levels in the catchment area. Groundwater levels typically rise during the winter season and decrease from April to September due to increased evapotranspiration during the

summer months. The model tends to overestimate groundwater levels at the downstream end of the catchment while underestimating them in the upstream region, specifically in the Belgium part. However, model performance varies considerably from one point to another. Owing to the shallow groundwater table, groundwater levels exhibit dynamic spatial and temporal responses to precipitation. Precipitation events directly influence local groundwater levels, resulting in varied groundwater performance across the area. The model's efficacy in predicting groundwater levels was evaluated using the



efficiency parameters R, NSE, and KGE, which ranged from 0.608 to 0.959, -0.241 to 0.842, and -0.298 to 0.856,
respectively. The variation of GWL for model calibration and validation period against observed data can be seen in Fig. S3
provided in supplementary material.

### 4.2.3 Catchment Water Balance

The overall water balance of the catchment exhibited consistent behaviour, characterized by precipitation as the primary
input and evapotranspiration as the major loss component. Streamflows were primarily influenced by the saturated zone
drain through the subsurface drainage system, followed by contributions from base flow and overland flow. Boundary flows
indicated a net outflow, indicating that more water left the catchment than entered it through the saturated zone. The areas at
the upstream and downstream ends of the catchment interacted with sources outside the aquifer, emphasizing the catchment's
subsurface hydrological connectivity.

### 4.2.4 Total Water Balance

The total water balance was assessed for both the calibration period (September 2009 to December 2016) and the validation
period (January 2017 to December 2021). Precipitation emerged as the key driver of the water balance, with approximately
60% of the total precipitation lost through evapotranspiration. The saturated zone component accounted for approximately
40% of the total precipitation, with 55% of that water contributing to surface water via the subsurface drainage system.
Throughout the entire simulation period, notable patterns were observed in the unsaturated zone (UZ) and saturated zone
(SZ) storage. The unsaturated zone storage exhibited a decline, while the saturated zone storage showed an increase. This
behaviour can be attributed to the relatively shallow unsaturated zone in the lowland catchment, resulting in higher levels of
evapotranspiration. Comprehensive overview of the contributions and proportions of the water balance components during
both the calibration and validation periods is provided in supplementary material (Table S5).

### 4.2.5 Wet and dry hydrological year water balance

The water balance of the catchment was examined separately for wet and dry hydrological years, which are defined as the
periods from October 1st to September 30th. Based on the analysis of precipitation and evapotranspiration data, the year
2010 was identified as a wet hydrological year, while 2018 was identified as a dry hydrological year. During the wet
hydrological year, approximately 51% of the total precipitation was lost through evapotranspiration, while 41% infiltrated
into the saturated zone (SZ). Out of this infiltrated amount, 39% drained back into the river as a base flow, resulting in
positive changes in SZ and unsaturated zone (UZ) storage. Conversely, in the dry hydrological year, around 70% of the
precipitation was lost through evapotranspiration, with only 32% infiltrating into the SZ. Among the infiltrated amount, 60%
drained into the river as a base flow, leading to negative changes in SZ and UZ storage. These findings emphasize that
during dry periods, the saturated zone/groundwater becomes the primary contributor to river flows. The detailed results of
the water balance analysis can be found in supplementary material (Table S6).



### 4.3 Precipitation spatial variability impact

The objective of this analysis was to examine the impact of spatial variability in precipitation data on the hydrology of a mesoscale lowland catchment. Three different representations of precipitation, namely Theisen polygon, IDW interpolation, and radar data, were tested in the fully distributed hydrological model MIKE SHE.

The study revealed that the influence of precipitation spatial variability on surface water components, specifically streamflow, was negligible. The average NSE computed for three streamflow measurement points displayed slight variations ranging from -1.9 % to 2.6%. However, adjustments in the spatial resolution of precipitation data did result in local fluctuations in groundwater levels. Notably, certain points exhibited a maximum increase of 168% and a decrease of 25% in NSE coefficients for groundwater levels. These observations can be attributed to the characteristics of the catchment. The Aa

of Weerijs catchment is relatively flat, and frontal precipitation predominates the region, resulting in a relatively uniform distribution of precipitation across the catchment. Consequently, modifying the resolution of precipitation data has minimal to no impact on streamflows. However, the spatial variability of precipitation within the catchment does affect local recharge, leading to observed variations in groundwater head at specific points. Fig 4 displays the variation in groundwater heads in terms of R and KGE, illustrating that the transition from uniform to spatially distributed precipitation representation

had limited and localized effects on groundwater levels throughout the catchment.

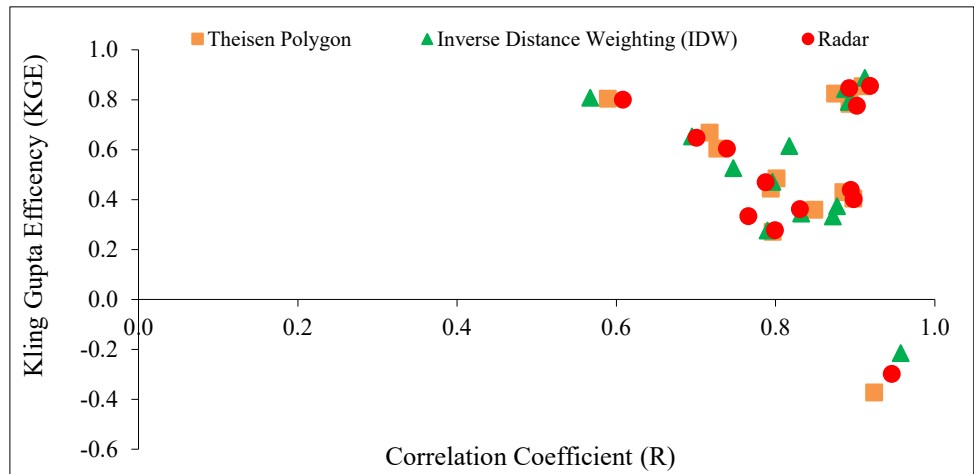

**Figure 4: Variation of GWL in terms of R and KGE for different precipitation representation**

The impact of precipitation spatial variability on the water balance of the catchment was found to be minimal. Nevertheless, the utilization of Radar data representation demonstrated an improved depiction of precipitation distribution across space.

This enhancement was particularly effective in capturing local variations in groundwater levels, providing a more accurate representation of the hydrological dynamics within the catchment. The detailed catchment water balance for the calibrated period under different precipitation representations is presented in supplementary material (Table S7).



## 5 Conclusions

A physically-based, fully-distributed hydrological model, MIKE SHE coupled with MIKE 11, was developed and applied to
the Aa of Weerijis catchment, which spans across the Netherlands and Belgium. The model was successfully calibrated
against three key variables: streamflow, GWL, and actual evapotranspiration, which were distributed throughout the
catchment. To ensure accurate calibration of the model against multivariate and multisite data, we implemented an optimized
weighted objective function using MR and MNSE. These objective functions allowed us to effectively capture the complex
relationships and variations in the observed variables throughout the catchment.  The model successfully captured the
hydrological dynamics of the catchment with a value of R as 0.9 at most of the observation points. Importantly, the model
accurately simulated the dynamic behaviour of GWL, influenced by local precipitation events and the shallow water table,
across multiple distributed points within the catchment. Precipitation was identified as the primary driver of catchment
hydrology, with substantial losses through evapotranspiration. Streamflows primarily originated from baseflow through the
subsurface drainage network. The catchment's shallow groundwater level facilitated intensive interactions between surface
water and groundwater, which were locally influenced by precipitation events. Understanding these dynamics is crucial for
effective water management strategies in the catchment. Furthermore, the impact of spatial variability in precipitation data
on catchment hydrology using the calibrated model was analysed. Three representations of precipitation distribution were
evaluated, ranging from a uniform distribution with Theisen polygons to fully-distributed approaches using IDW and radar
data.  Research findings revealed that altering the spatial resolution of precipitation had no significant effect on streamflow.
However, local variations in groundwater heads were observed due to slight spatial variations in precipitation. The Aa of
Weerijs catchment is characterized by flat topography, and frontal precipitation is the primary meteorological feature in the
region. This results in a relatively uniform distribution of precipitation throughout the catchment. Therefore, any changes in
the spatial variability of precipitation do not significantly affect the total amount of precipitation, leading to minimal impact
on the catchment's hydrology. Moreover, the final calibrated version of the developed distributed hydrological model for the
Aa of Weerijs catchment holds practical value for future water resource planning to mitigate the impacts of climate change.

## Authors contribution

F.S., M.H.A., I.P. A.J., S.J.V.A. and C.B. conceptualized and designed the study. F.S. and M.H.A. undertook the simulation
and analyses of results. F.S. wrote the manuscript and figure preparation. Final review and edits were done by M.H.A. and
I.P.



## Acknowledgements

Research presented here was supported by the European Union's Horizon 2020 research and innovation programme "EIFFEL project" (No.101003518). We thank Waterboard Brabantse Delta, Breda, the Netherlands, for data provision and support in this research.

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
