# Peer review of "Surface-subsurface interaction analysis and the influence of precipitation spatial variability on a lowland mesoscale catchment"

_Hydrology and Earth System Sciences, 2023_

## Referee Comment (RC1)

I am sorry. I am sure I will reject this manuscript after I finish reading the abstract and I cannot continue after I read the first paragraph of introduction.

Abstract:
lines 14-16: authors said that they used the NSE and R, so what are the values they got? As opposed to listing the values, authors said 'model performed well, with satisfactory….', so what do 'well' and 'satisfactory' mean?

The only description of interactions between groundwater and surface water is in line 18, a very vague expression, but authors said, 'the findings highlight the importance of surface water-groundwater interactions ….' in line 21. Why?

Overall, all the conclusions in abstract are fundamental knowledge of hydrology in textbooks.

Introduction:
It is very vague. for example, line 26, 'a multi-scale approach that combines various techniques', what do the authors want to express? I got nothing.

Then lines 28–31, some other fundamental knowledge. Also, I don't think 'groundwater tables' is a right terminology, it should be 'water table'.

Then lines 32-33, what does 'unique characteristics' mean? I got nothing again.

why 'the temporal variations' in line 33? why not spatial?

lines 34-35, the cited papers are 15-20 years ago. why 'empirical field-based and numerical modeling', why not remote sensing and machine learning?

Then from line 36 to the end of the introduction, I got nothing new and the only useful information is a modeling work was conducted in the catchment, Aa of Weerijs, based on MIKE SHE and MIKE 11. Another interesting thing is, in line 79, authors said it is a mesoscale catchment while in line 86, it becomes a small scale.

lines 102–103, 'The hydrology of the Aa of Weerijs catchment is influenced by various factors, including topography, land use, soil type, and climate.' I am wondering which catchment is not affected by various factors as listed by authors?

lines 112–121, always, in this part, the governing equations and coupling approach are introduced. However, I got nothing again in this part. Though I knew MIKE SHE and MIKE 11 well. But if I was a common reader, I do have a question 'what MIKE SHE and MIKE 11 are?' I can only get that they are powerful and they can simulate everything, then what?

For the model itself, nothing is attractive. 500 resolution modeling over a domain of 346 km$^2$

I am really sorry. I suggest authors submit it to another journal. This is good documentation about the study of MIKE model in a real catchment. I didn't find any advanced scientific question and anything different in terms of model and modeling. I really don't think HESS is the right place.

---

## Referee Comment (RC2)

Review for manuscript: *Surface-subsurface interaction analysis and the influence of precipitation spatial variability on a lowland mesoscale catchment*

**Comment:**

After reviewing the manuscript, several critical points emerged that raise concerns regarding the scientific rigor and presentation quality. Here are some specific areas of concern:

1. The abstract fails to adequately address key factors influencing the conceptual model, such as the regulation of streamflow and the agricultural nature of the watershed. These factors are crucial for understanding the context and conditioning of the modeling processes, yet their omission diminishes the clarity and completeness of the manuscript.

2. Acronyms, including MIKE SHE and MIKE 11, are frequently used throughout the manuscript without proper explanation or clarification of their respective roles and differences. This lack of clarity hinders readers' understanding of the modeling approach and its components.

3. The manuscript contains numerous instances of vague language lacking both literature and quantitative support. Additionally, figures and tables are inadequately labeled, with critical information missing, such as a color scale for Figure 3. Confusing equation notations (Equation 1 and 2) and setups (line 201-202) further complicate comprehension and interpretation.

4. Much of the modeling process appears to be subjective, with weights assigned based on authors' knowledge (line 203-209) and parameter tuning conducted manually (line 211-212) for purported insights. This subjectivity raises questions about the validity and reliability of the results, particularly considering the authors' acknowledgment that "promising" results (line 252-254 ) align with their mental model, suggesting potential bias and the generation of artifacts.